# Synthesis and Antimicrobial Activity of Metal-Containing Linseed Oil-Based Waterborne Urethane Oil Wood Coatings

**DOI:** 10.3390/polym12030663

**Published:** 2020-03-16

**Authors:** Kun-Tsung Lu, Jing-Ping Chang

**Affiliations:** Department of Forestry, National Chung Hsing University, 250, Kuo-Kuang Rd., Taichung 402, Taiwan; guluwalala@gmail.com

**Keywords:** linseed oil, waterborne urethane oil, mono(hydroxyethoxyethyl) phthalate, antimicrobial activity, wood coatings

## Abstract

In this study, the antimicrobial agents of mono(hydroxyethoxyethyl)phthalate (M(HEEP)_2_) with different metal of M = Zn, Mn, Pb, and Ca were synthesized from diethylene glycol (DEG), phthalic anhydride (PA), and divalent metal acetates including calcium acetate, zinc acetate, manganese acetate, and lead acetate, respectively. The waterborne urethane oil (WUO) dispersions synthesized from linseed oil, diisocyanates (hexamethylene diisocyanate (HDI) and isophorone diisocyanate (IPDI)), dimethylolpropionic acid at NCO/OH molars of 0.9, by acetone processing method were described as in our previous report. The M(HEEP)_2_ antimicrobial agents as well as the commercial nanosilver powder were added into WUO dispersions as the antimicrobial coatings. The effects of various antimicrobial agents and dosages (0.0, 0.2, 0.6, 0.8, 1.0, 2.0, and 4.0 phr) on antimicrobial activity of WUO films against gram-negative bacterium of *Escherichia coli*, gram-positive bacterium of *Staphylococcus aureus*, brown-rot fungus of *Gloeophyllum trabeum*, and white-rot fungus of *Lenzites betulina* were assessed. In addition, the film properties of the best antimicrobial WUO coatings were also examined. The results showed that the antimicrobial agents of mono(hydroxyethoxyethyl) phthalate M(HEEP)_2_ (M = Zn, Mn, Pb, and Ca) powders should certainly be synthesized by FTIR, ^1^H-NMR, ^13^C-NMR, and energy-dispersive X-ray spectroscopy (EDS) identifications and the yields of them were 43–55%. The results also revealed that the WUO film synthesizing with HDI films containing Zn(HEEP)_2_ of 2.0 phr and Pb(HEEP)_2_ of 0.4 phr had the best antibacterial activity for *E. coli* and *S. aureus*, respectively. The IPDI films containing Zn(HEEP)_2_ of 1.0 phr had the best antibacterial activity for both *E. coli* and *S. aureus.* For antifungal activity, the WUO film synthesizing with HDI films containing Pb(HEEP)_2_ of 0.8 phr and Zn(HEEP)_2_ of 2.0 phr as well as IPDI films containing Mn(HEEP)_2_ of 0.2 phr and Zn(HEEP)_2_ of 4.0 phr had the best performances against *G. trabeum* and *L. betulina*, respectively. Comparing with commercial nanoAg powder, the Zn(HEEP)_2_ and Pb(HEEP)_2_ had a superior antifungal efficiency for *G. trabeum* and *L. betulina*, while it had a slightly inferior efficiency in the antibacterial activity for *E. coli* and *S. aureus*. On the properties of WUO films, adding metal-containing antimicrobial agents could slightly enhance the thermal stability, but lowered the gloss of all films, however, the *T*_g_ value increased for HDI film and decreased for IPDI film. In addition to this, they had no significant difference in the film properties including hardness, impact resistance, bending resistance, adhesion, mass retention, and light-fastness between the WUO films with and without adding antimicrobial agents.

## 1. Introduction

The dwindling petroleum supplies have enormously increased the material cost as well as deep environmental concerns. In order to reduce the reliance on petrochemicals, employing renewable resources, such as natural linseed oil, tung oil, or castor oil, to replace petroleum derivatives for manufacturing wood coatings [1,2,3,4,5,6] and currently, waterborne coatings have received more attention as a substitute for solvent-borne coatings due to their nontoxic and nonflammable character as well as the emission of lesser or no volatile organic compounds (VOCs) [7,8]. In addition, the safety, healthy, ecology, and amenity living environments are the goal for human life [2,9,10]. Consequently, introducing the antimicrobial function on the articles for daily use such as wood furniture and building materials will decrease significantly the pollution of bacteria and fungi. Zafar et al. [11] reported that zinc incorporated linseed oil based polyesteramide was successfully synthesized at a lower temperature without an organic solvent. It was found that minor incorporation of zinc in linseed oil based polyesteramide exhibited improved antibacterial activities against *Escherichia coli* and *Staphylococcus aureus*. Zafar et al. [12] also founded that zinc and cadmium incorporated linseed oil based poly(esteramide-urethane) (Zn/Cd-LPEAUr) were performed against *E. coli* and *S. aureus*, and compared with those of petroleum-based metal containing polyurethanes. Zafar et al. [13,14] further reported that Zn-containing self-cured *Pongamia glabra* oil based polyesteramide (Zn-APGPEA) resin and linseed oil based metallopolyesteramides (Mn(II)-/Co(II)-/Cu(II)-LPEA) containing metals resins can be used as an excellent antibacterial performance material against *E. coli* and *S. aureus*. Hsu et al. [15] prepared nanocomposites from a polyester-type waterborne polyurethane (PU) containing various low concentrations of silver nanoparticles (nanoAg) and found that nanoAg could not only improve the physical properties and biocompatibility of PU, but also inhibit the growth of bacteria of *Bacillus subtilis* and *E. coli*. Li et al. [16] found that PU films doped with ZnO nanoparticles showed excellent antibacterial activity, especially for *E. coli*. Jayakumar et al. [17,18,19] prepared the metal salts of mono(hydroxyethoxyethyl)phthalate by reacting phthalic anhydride with diethylene glycol and metal acetate, where metal is Cu^2+^, Mn^2+^, and Zn^2+^. The metal-containing polyurethanes in the main chain were further synthesized by reacting hexamethylene diisocyanate (HMDI) or toluene 2,4-diisocyanate (TDI) with the salts of mono(hydroxyethoxyethyl)phthalate. It also found that these metal-containing polyurethanes had antibacterial activity against *E. coli*, *Pseudomonas fluorescence*, *Streptococcus* sp., and *Salmonella* sp. From the reports mentioned above, exhibiting the metal-containing materials possess the antimicrobial activity. However, in our pre-experiments on the mono(hydroxyethoxyethyl)phthalate reacting with diisocyanates, the polyurethanes materials would be gelled. The results were also confirmed by Matsuda [20], who reported that the polyurethane obtained were glassy materials or white powders. It is difficult to carry out in practical wood finishing. Therefore, in this study the mono(hydroxyethoxyethyl) phthalate were synthesized alone and the powders were used as additives or fillers for antimicrobial agents.

In this study, the waterborne urethane oil (WUO) dispersions synthesized from linseed oil, diisocyanates (hexamethylene diisocyanate and isophorone diisocyanate), dimethylolpropionic acid at various NCO/OH molars of 0.7, 0.8, and 0.9, are described as in our previous report [21]. In addition, the films properties of the WUOs for wood coatings are listed in another report [22]. The results showed that all of the WUO films had excellent adhesion, durability, and lightfastness, especially the HDI-0.9 (hexamethylene diisocyanate and NCO/OH molars of 0.9) and IPDI-0.9 (isophorone diisocyanate and NCO/OH molars of 0.9) have balance properties between the coating and film of WUO, and are suitable for an oil finishing wood coating. Therefore, the linseed oil-based waterborne urethane oil wood coatings, HDI-0.9 and IPDI-0.9, were used in this study. Furthermore, the antimicrobial agents of mono(hydroxyethoxyethyl)phthalate (M(HEEP)_2_) with a different metal of M = Zn, Mn, Pb, and Ca were also synthesized. The different types and various dosages of antimicrobial agents as well as the commercial nanoAg powder, which is known to have antimicrobial properties [15] as a control group were added into WUOs as antimicrobial wood coatings for a finishing on furniture and building materials. The antimicrobial activity of WUO films against Gram-negative bacterium of *Escherichia coli*, Gram-positive bacterium of *Staphylococcus aureus*, brown-rot fungus of *Gloeophyllum trabeum,* and white-rot fungus of *Lenzites betulina* were assessed. In addition, the film properties of the best antimicrobial wood coatings were also examined.

## 2. Materials and Methods

### 2.1. Materials

Diethylene glycol (DEG), divalent metal salts including calcium acetate, zinc acetate, manganese acetate, and lead acetate, which are hydrated, were purchased from Choneye Pure Chemicals Co. Ltd. (Taichung, Taiwan). Phthalic anhydride (PA) was purchased from Shimakyu’s Pure Chemicals Co. Ltd. (Osaka, Japan). Commercial nanosilver powder (nanoAg), which is a composite with zirconium phosphate and the nanosilver content was 6% and was purchased from Chin-Tai Resins Chemical Co. Ltd. (Taichung, Taiwan). Methanol was purchased from Aencore Chemical Co. (Surrey Hills, Australia). Acetone was purchased from Union Chemical Works Ltd. (Taichung, Taiwan). Deionized water was prepared in our laboratory. A waterborne metal dryer including cobalt (Co-dryer, metal content of 5 wt %), zirconium (Zr-dryer, metal content of 12 wt %), and calcium (Ca-dryer, metal content of 5 wt %) dryers were supplied by Qing Yi Co. Ltd. (Taichung, Taiwan). A filter paper disc was obtained from Toyo Roshi Kaisha, Ltd. (Bunkyo-ku, Japan). Culture media of potato dextrose agar (PDA) was purchased from Merck Taiwan Ltd. (Taipei, Taiwan). Microbial strains including *Escherichia coli* (Gram-negative bacterium), *Staphylococcus aureus* (Gram-positive bacterium), *Gloeophyllum trabeum* (brown-rot fungus), and *Lenzites betulina* (white-rot fungus) were purchased from the Food Industry Research and Development Institute (Hsinchu, Taiwan). *Cryptomeria japonica* boards (with a moisture content of 10.7%), glass sheets, tin-coated iron sheets, and Teflon sheets were all prepared as specified by the CNS 9007 Standard [23] and were used as experimental substrates for film properties testing.

### 2.2. Preparation of Waterborne Urethane Oil (WUO)

The waterborne urethane oil (WUO) dispersions synthesized from linseed oil, diisocyanates (hexamethylene diisocyanate (HDI) and isophorone diisocyanate (IPDI)), and dimethylolpropionic acid at NCO/OH molars of 0.9 by the acetone processing method were described as in our previous report [21]. The linseed oil glyceride (LOG) was first synthesized by using a transesterification process with a glycerol/linseed oil molar ratio of 1.0. Then the dimethylolpropionic acid reacted with HDI or IPDI, followed by adding LOG at NCO/OH molars of 0.9, and the COOH-containing prepolymer was obtained. Then, the ionomer, which was prepared by a neutralizing prepolymer with trimethylamine, was dispersed by adding deionized water, and the water–acetone dispersion was obtained. Finally, the acetone was removed by vacuum distillation and the WUO were obtained, respectively, and named the HDI-0.9 coating and IPDI-0.9 coating.

### 2.3. Synthesis and Identification of Metal Containing Mono(hydroxyethoxyethyl) Phthalate (M(HEEP)_2_)

The synthesis method referred Jayakurmar et al. [17] and Matsuda [20] reports and we modified the process as follows. The 4 mole diethylene glycol (DEG) and 1 mole phthalic anhydride (PA) were placed in a four-neck reaction flask. The mixture was stirred and temperature rose up to 135 °C and maintained for 1.5 h. Then 0.5 mole divalent metal acetates (metal (M) = Ca, Zn, Mn, and Pb) were added respectively and the reaction temperature of the mixtures was adjusted to a specific temperature, i.e., 162 °C for calcium acetate, 150 °C for zinc acetate, 140 °C for manganese acetate, and 135 °C for lead acetate. When the specific temperature reached, the mixture was continued stirred at the temperature for another 3 h. During the synthesis process, the appearance of the mixture transferred to milk white color and insoluble precipitates produced. Subsequently, the reaction system was cooled to room temperature and using acetone, methanol, and xylene washed the insoluble precipitates repeatedly. The powder of the metal containing mono(hydroxyethoxyethyl) phthalate (M(HEEP)2) was obtained and oven dried at 60 °C for 3 days. The structure, appearance, and constituents of the powders were identified by Fourier-transform infrared spectroscopy (FTIR; Perkin-Elmer Spectrum 100 spectrometer using ATR mode; PerkinElmer, Waltham, MA, USA), nuclear magnetic resonance (NMR; Aglient Technologies DD2-600Hz, including 1H-NMR and 13C-NMR; Agilent Technologies, CA, USA), field emission scanning electron microscopes (FE-SEM; JEOL JSM-6700F; Japan Electron Optics Laboratory Co., Ltd., Tokyo, Japan), and energy-dispersive X-ray spectroscopy (EDS; Oxford X-Act 10 mm2, Oxford Instruments, Abingdon, Oxfordshire, England), respectively.

### 2.4. Determination of Antimicrobial Activity

Antibacterial and antifungal activities of M(HEEP)_2_-containing WUO coatings were investigated by the agar disk diffusion method [24,25]. A 39 g/L of potato dextrose agar (PDA) sterilized in an autoclave sterilizer as a culture medium. The 0.1 mL of prepared culture suspension (ca. 10^7^–10^8^ cfu/mL) was spread on the solidification surfaces of the agar culture medium (20 mL) on a petri dish. The dosages of various M(HEEP)_2_, M = Ca, Zn, Mn, and Pb and commercial nanoAg powders were 0.2, 0.4, 0.6, 0.8, 1.0, 2.0, and 4.0 phr according to the solid content of the WUO coating and the mixtures were shaken by ultrasonic waves for 1 h. Then the filter paper disks (8 mm in diameter) were first impregnated with WUO coatings for 5 min, followed by being air dried for 3 days and then sterilized by UV radiation for 8 h. The sterile filter paper disks were put on agar surfaces. The zones of growth inhibition around the disks were measured after 16–24 h of incubation at 37 °C for bacteria and 3 days for fungi at 25 °C. According to the sizes of the inhibitory zone (including the diameter of disk) on the agar surface around the disks, the antimicrobial activity was graded as: - inactive (0–9 mm); + mildly active (10–15 mm); ++ moderately active (16–20 mm); and +++ highly active (≥21 mm), respectively [18,19]. An average value and standard deviation of triplicate measurements was reported.

### 2.5. Determination of the Film Properties of WUO Coatings

The best antimicrobial activity of M(HEEP)_2_-containing the WUO coating was chosen to determine the film properties and the commercial nanoAg-containing one was used as a control group. The 0.3 wt % of the waterborne metal dryer including cobalt, zirconium, and calcium dryers were added to the WUO coating respectively, by the weight of the solid content of the coating and stirred for 30 min. The selected substrates were coated using a universal applicator with a wet film thickness of 250 μm, and were placed in a constant temperature for 10 min and then on the oven at 50 °C for another 10 min, followed by drying in an air-conditioned environment at 26 °C and 60% RH for 30 days. The tests were carried out after this drying process.

The hardness of tested film on glass sheets was conducted using a König/Persoz Pendulum Hardness Tester (Braive Instruments, Liège, Belgium), according to the ISO 1522 [26]. Ten points were tested with the values averaged for each specimen. Impact resistance of films on wood was investigated based on the falling weight was 300 g and an impact needle diameter of 2.54 cm by using the Dupont Impact Tester IM-601 The bending resistance of tested films on tin-coated iron sheets was carried out according to JIS-K-5400 [27] by using a bending tester (Ueshima Seisakusho Co., Ltd., Tokyo, Japan) with steel bars diameters of 2, 3, 4, 6, 8, and 10 mm. The adhesion of tested films on wood was performed by using the crosscut method according to CNS 10756 K 6800 [28]. The best adhesion is Grade 10 followed by Grades 8, 6, 4, 2, and 0. The mass retention of tested films, which were first coated on Teflon sheets and then were separated off as a free film, was determined by putting each given weighted films into a Soxhelt extractor (Dogger Co., New Taipei City, Taiwan) containing 250 mL acetone. The solution was siphoned four times per hour (total 6 h), and the soaked film was dried in an oven at 50 °C for 6 h and the weight retention was calculated. The gloss of films coated on wood panels and parallel to wood grain was detected by using a Dr. Lange 60° Reflectometer (Dr. Bruno Lange GmbH, Berlin, Germany). The tests mentioned above were performed at least triplicate and averages were recorded.

The lightfastness of films coated on white card paper was carried out with a Paint Coating Fade Meter (Suga Test Instruments, Japan). The light source was mercury light (H400-F), and chamber temperature was 32 ± 4 °C. After a given exposure time, the changes in color of the specimens were measured with a spectrophotometer (CM-3600d, Minolta. Osaka, Japan) with an 8 mm target mask and fitted with a D65 light source with a measuring angle of 10°. The tristimulus values X, Y, and Z of all specimens were obtained directly from the colorimeter. The CIE L*, a*, and b* color parameters were then computed, followed by calculating the color difference (ΔE*). Dynamic mechanical analysis (DMA) of the films, which were prepared the same as the mass retention tested film, was to determine the glass transition temperature (*T*_g_) based on a single mode using a PerkinElmer DMA 8000 (PerkinElmer, Waltham, MA, USA). The test was performed in a nitrogen atmosphere and the resonance frequency was adjusted to 1 Hz. with the temperature increasing from −50 to 150 °C at a heating rate of 2 °C min^−1^. The thermogravimetric analysis (TGA) of the films, which were prepared the same as mass retention tested film, was conducted using a PerkinElmer STA 6000 (PerkinElmer, Waltham, MA, USA) in a nitrogen atmosphere with the temperature increasing from 50 to 700 °C at a heating rate of 10 °C min^−1^.

## 3. Results and Discussion

### 3.1. Synthesis and Identification of Metal Containing Mono(hydroxyethoxyethyl) Phthalate (M(HEEP)_2_)

In our pre-experiment, the metal containing mono(hydroxyethoxyethyl) phthalate (M(HEEP)_2_) was synthesized from diethyl glycol (DEG), phthalate (PA), and various of metal acetates including Zn, Mn, Pb, and Ca acetates at 60–70 °C as described in Matsuda [20] and Jayakurmar et al. [17], however, the yields of M(HEEP)_2_ were only 4–5%. In our modified process, the mixture of PA and DEG was stirred and temperature was rose up to 135 °C, which was higher than the melt point of PA of 131 °C, and maintained for 1.5 h to obtain mono diglycolic phthalate. Then the divalent metal acetates were added respectively, and the reaction temperature of the mixtures was adjusted to a specific temperature (as listed in Table 1) according to the insoluble precipitates produced. When the specific temperature reached, the mixture was continued stirred at the temperature for another 3 h. The results showed that the yields of the M(HEEP)_2_ were increased to 43.4–55.1%. The appearance of the M(HEEP)_2_ powder was a milk white color except the Mn(HEEP)_2_ powder was a light pink color as shown in Figure 1.

The FTIR spectra of M(HEEP)_2_ are shown in Figure 2. A broad absorption peak at 3400–3500 cm^−1^, which indicates the presence of OH groups and the peak that represents the carbonyl (C=O) group stretching vibration was observed at 1700–1730 cm^−1^. The peaks at 1400–1430 and 1545–1570 cm^−1^ indicate the ion bonding vibrations between carbonyl group and metal ions. In addition, peaks at 1080–1130 cm^−1^ (C–O–C stretching vibration), 1045–1060 cm^−1^ (primary alcohol of C–O stretching vibration), and 725–735 cm^−1^ (C–H out of plane bending vibration of the benzene ring) [11,13,17,18,19,20] indicates that M(HEEP)_2_ was readily synthesized.

The structures of M(HEEP)_2_ were also investigated by NMR spectrometry. Taking Ca(HEEP)_2_ as an example, the ^1^H-NMR and ^13^C-NMR spectra are displayed by Figure 3 and Figure 4, respectively. In the ^1^H-NMR spectrum, the signals were observed in the 7.41–7.48 ppm (H1), which corresponded to the aromatic proton. The signals with the chemical shift at 4.21–4.25 ppm (H2) could be assigned to the proton of –COOC**H**_2_–, and the proton of –O**H** shows signals at 3.86–3.88 ppm. (H3) The peaks at the 3.71–3.76 ppm (H4) region could be attributed to the methylene group proton of R–COOAr, and the protons of –C**H**_2_–O–C**H**_2_– show signals at 3.61–3.64 ppm (H5) [11,13,17,18,19,20].

The ^13^C-NMR spectrum of Ca(HEEP)_2_ (Figure 4) shows the carbon of COO– bonding with Ca (C12) and COO– adjacent to CH_2_ (C5) at 176.7–177.1 ppm and 169.3 ppm, respectively. The signal at 136.2 ppm (C11) and 132.5 ppm (C6) assigned to the carbon atoms of the aromatic group attached to COO–. The aromatic carbons C7, C8, C9, and C10 appeared in the 126.6–129.3 ppm regions. The carbon of the methylene shows a peak at 71.5 ppm (C1) for –**C**H_2_–OH, at 71.7 ppm (C4) for –**C**H_2_OOC–Ar, at 60.3 ppm (C2) for –**C**H_2_–CH_2_OH, and 68.2 ppm (C3) for –**C**H_2_–CH_2_OOC–Ar [11,13,17,18,19,20]. These results indicated clearly that the Ca(HEEP)_2_ was synthesized successfully from DEG, PA, and calcium acetate.

The morphologic characterization of M(HEEP)_2_ was determined by the FE-SEM analysis as listed in Figure 5. The Zn(HEEP)_2_ was piled as a sheet-like shape and had the largest size; the Ca(HEEP)_2_ was piled as a columnar shape; the Mn(HEEP)_2_ was piled as a spherical shape; and the Pb(HEEP)_2_ was piled as a lumpy shape and had the smallest size. The different morphology and size of M(HEEP)_2_ crystals may be due to the different nucleation during the synthesis process [29]. The melt points of zinc acetate (200 °C) and calcium acetate (160°C) were higher than or equal to the synthesis temperature (as shown in Table 1), resulting in an uneven nucleation and obtaining a larger size crystal. On the contrary, the manganese acetate and lead acetate had a lower melt point of 80 and 75 °C, which was also lower than the synthesis temperature, attributing to an even nucleation and a smaller size crystal obtained. In addition, the EDS was used for confirming the metal ions were introduced into the mono(hydroxyethoxyethyl) phthalate as displayed by Figure 6. The results further revealed that the M(HEEP)_2_ (M = Zn, Mn, Pb, and Ca) were synthesized successfully as shown in Scheme 1.

### 3.2. Antibacterial Activity of M(HEEP)_2_-Containing WUO Coatings

In this study, the WUO coatings with hexamethylene diisocyanate (HDI) and isophorone diisocyanate (IPDI), and dimethylolpropionic acid at NCO/OH molars of 0.9 were used, which were, respectively, named the HDI-0.9 coating and IPDI-0.9 coating. The bacteria including *E. coli* (Gram-negative bacterium) and *S. aureus* (Gram-positive bacterium) were used and the agar disk diffusion method was applied for determining the antibacterial activity. The antibacterial activity of various M(HEEP)_2_ (M = Zn, Mn, Pb, and Ca) and commercial nanoAg dosages containing WUO films synthesizing with HDI and IPDI for *E. coli* are listed in Table 2 and Table 3, respectively, and the inhibitory zone of Zn(HEEP)_2_ for *E. coli* that was used as an example is presented in Figure 7. The results showed that the inhibitory zone of the blank group (WUO films containing 0 phr Zn(HEEP)_2_)) was 8 mm (Figure 7a), meaning inactive on antibacterial activity, while the inhibitory zone increased to 15 mm (as shown in Figure 7b) with increasing Zn(HEEP)_2_ dosage of 2.0 phr, assigning mildly active antibacterial activity. However, the inhibitory zone decreased to 11 mm with increasing the Zn(HEEP)_2_ content to 4.0 phr (Figure 7c). The Table 2 showed that in the HDI-0.9 coating, except the Zn(HEEP)_2_ must add to 0.6 phr, the other M(HEEP)_2_ ((M = Mn, Pb, and Ca) and commercial nanoAg had only the dosage of 0.2 phr had promptly a mildly active antibacterial activity for *E. coli*.

Table 3 for IPDI-0.9 coatings, the inhibitory zone of without M(HEEP)_2_ was 8 mm, indicating inactive antibacterial activity for *E. coli*. The Zn(HEEP)_2_ containing WUO film had a superior antibacterial activity, its inhibitory zones increased from 11 to 14 mm with an increasing dosage of 0.2–1.0 phr. The inhibitory zones of Mn(HEEP)_2_ containing WUO films were 10–11 mm, also assigning a mildly active, but they show less efficiency on antibacterial activity than the Zn(HEEP)_2_ containing WUO films. In addition, the Pb(HEEP)_2_ and Ca(HEEP)_2_ containing WUO films (IPDI-Pb and IPDI-Ca) had inactive antibacterial activity for *E. coli*. However, the control group of commercial nanoAg containing WUO films (IPDI-Ag) possessed inhibitory zones of 19–23 mm and had the highest antibacterial activity for *E. coli*. Even though by only adding 0.2 phr of nanoAg, it shows a highly active antibacterial activity.

The antibacterial activity of various M(HEEP)_2_ (M = Zn, Mn, Pb, and Ca) and commercial nanoAg dosage containing WUO films synthesizing with HDI and IPDI for *S. aureus* are listed in Table 4 and Table 5, respectively. The results showed that the inhibitory zone of WUO films without M(HEEP)_2_ and commercial nanoAg was 9 mm, meaning inactive antibacterial activity for *S. aureus*. The inhibitory zone of HDI-Zn, HDI-Mn, and HDI-Ca films were 11–14 mm, 12–14 mm, and 13–15 mm, respectively, assigning mildly active antibacterial activity. However, the HDI-Pb films had a larger inhibitory zone of 15–16 mm, especially the ones of 0.4–1.0 phr possessed a superior antibacterial activity, which was compared with the control group of HDI-Ag films, which the inhibitory zone was 18–19 mm and revealed a moderately active antibacterial activity for *S. aureus*.

Table 5 shows the IPDI-0.9 coatings on antibacterial activity for *S. aureus*. The inhibitory zone of IPDI-Mn, IPDI-Pb, and IPDI-Ca were 12–14 mm, 12–14mm, and 11–13 mm, respectively, indicating a mildly active on antibacterial activity for *S. aureus*. However, the Zn(HEEP)_2_ containing WUO films had a larger inhibitory zone of 15–19 mm, meaning a superior moderately active antibacterial activity, which was slightly less or equal to that of commercial nanoAg containing WUO films, which possessed a highly active antibacterial activity with a dosage of 0.8, 1.0, and 4.0 phr.

Integrating the results mentioned above, it could be concluded that except the IPDI-Pb and IPDI-Ca for *E. coli,* all of the WUO films with M(HEEP)_2_ had antibacterial activity for *E. coli* and *S. aureus*. Furthermore, the antibacterial efficiency for *S. aureus* was higher than that for *E. coli.* This might be due to the *S. aureus* belonging to Gram-positive bacterium, which the cell wall is composed of thickening peptidoglycan and takes a negative charge on the surface, which is more sensitivity to the metal ion with a positive charge than the Gram-negative bacterium of *E. coli.* In addition, the antibacterial efficiency of IPDI-Ag films was superior to that of HDI-Ag, which might be attributed to the IPDI-0.9 coating having a larger particle size of the z-average diameter of 995 nm than the HDI-0.9 coating of 719 nm [21], and the nanoAg was easy and well dispersed in the waterborne IPDI-0.9 coating, on the contrary it was easy agglutinated in the HDI-0.9 coating, resulting in a less free Ag ion and leading to inferior antibacterial activity.

### 3.3. Antifungal Activity of M(HEEP)_2_-Containing WUO Coatings

In this study, the fungi including brown-rot fungus of *G. trabeum* and white-rot fungus of *L. betulina* were used and the agar disk diffusion method was applied for determining the antifungal activity. The antifungal activity of various M(HEEP)_2_ (M = Zn, Mn, Pb, and Ca) and commercial nanoAg dosages containing WUO films synthesizing with HDI and IPDI for *G. trabeum* are listed in Table 6 and Table 7, respectively. The results showed that the inhibitory zone of blank group (0 phr) was 8 mm, meaning inactive antifungal activity for *G. trabeum.* The WUO films with HDI and containing M(HEEP)_2_ including HDI-Zn, HDI-Mn, HDI-Pb, HDI-Ca, and commercial nanoAg (HDI-Ag) had the inhibitory zone of 11–13 mm, 9–12 mm, 11–13 mm, 9–11 mm, and 10–11 mm, respectively, revealing only a dosage of 0.2 phr had mildly active antifungal activity for *G. trabeum.* Furthermore, even though the HDI-0.9 films with antimicrobial agents attributed to mildly active antifungal activity, the HDI-Zn and HDI-Pb exhibiting a superior antifungal efficiency compared to HDI-Ag.

The IPDI-0.9 films with antimicrobial agents (as shown in Table 7) had similar results to the HDI-0.9 films, especially the IPDI-Zn, IPDI-Mn, and IPDI-Ca with a dosage of 0.2 phr had a higher antifungal activity for *G. trabeum* than IPDI-Ag and that of HDI-0.9 films. In particular the inhibitory zone of IPDI-Mn was 16 mm, assigning a moderately active grade.

The antifungal activity of various M(HEEP)_2_ (M = Zn, Mn, Pb, and Ca) and commercial nanoAg dosages containing WUO films synthesizing with HDI and IPDI for white-rot fungus of *L. betulina* are listed in Table 8 and Table 9, respectively. The results indicated that among all of the WUO films, the HDI-Zn had the best antifungal activity for *L. betulina*, but the dosage of Zn(HEEP)_2_ must exceed 1.0 phr. In addition to this, the HDI-Mn, HDI-Pb, HDI-Ca, and control HDI-Ag had almost inactive antifungal activity for *L. betulina.* The IPDI-0.9 films had similar results to HDI-0.9 films, only the IPDI-Zn with a dosage of 4.0 phr had an inhibitory zone of 13 mm, assigning a mildly active antifungal activity.

From the results of antimicrobial activity, it can be seen that the metal-containing WUO films had more efficiency on antibacterial activity for bacterial than on antifungal activity for fungi. This may be attributed to the metal ions that can directly and easily damage the bacterial cell wall, by release of ions followed by an increased membrane permeability, loss of the proton motive force, and efflux of intracellular components [30]. The mechanism of metal ions on inhibiting microbial growth is an interesting topic and will be studied in the future.

### 3.4. Film Properties of WUO Coatings

In the present study, the M(HEEP)_2_ (M = Zn, Mn, Pb, and Ca) were additives for antimicrobial agents. Even though the results revealed that the WUO film synthesizing with HDI films containing Zn(HEEP)_2_ of 2.0 phr and Pb(HEEP)_2_ of 0.4 phr had the best antibacterial activity for *E. coli* and *S. aureus*, respectively. The IPDI films containing Zn(HEEP)_2_ of 1.0 phr had the best antibacterial activity for both *E. coli* and *S. aureus.* For antifungal activity, the WUO film synthesizing with HDI films containing Pb(HEEP)_2_ of 0.8 phr and Zn(HEEP)_2_ of 2.0 phr as well as IPDI films containing Mn(HEEP)_2_ of 0.2 phr and Zn(HEEP)_2_ of 4.0 phr had the best performances against *G. trabeum* and *L. betulina*, respectively. From the results mentioned above it also shows that the WUO films of HDI-Pb and IPDI-Zn, which were synthesized with HDI and IPDI, respectively, had the best antimicrobial efficiency for bacteria and fungi used in the study. According to the antimicrobial activity grades, the dosage of Zn(HEEP)_2_ and Pb(HEEP)_2_ was only 0.2 phr, which exhibited superiority of antimicrobial activity. Therefore, in this section, the films properties of HDI-0.9 (0.0 phr), HDI-Pb (0.2 phr), HDI-Ag (0.2 phr as control group), IPDI-0.9, IPDI-Zn, and IPDI-Ag were compared. The results are listed in Table 10.

The WUO film of IPDI-0.9 had a higher hardness of 87 sec than HDI-0.9 of 13 sec, while the hardness of metal-containing WUO films decreased slightly by adding the Zn(HEEP)_2_, Pb(HEEP)_2_, and nanoAg, which was from 13 to 11 sec for HDI films and from 87 to 72–76 sec for IPDI films. The IPDI film had a higher hardness than the HDI film. The results were also confirmed by the studies of Chang et al. [22]. The impact resistance of all HDI films was 25 cm, but of all IPDI films was lower than 5 cm. The bending resistance of all HDI films was lower than 2 mm-diameters and was superior to all IPDI films of 10 mm-diameter steel shaft. The results demonstrated that there had no difference between with or without metal-containing and kinds of metal ions on the impact and bending resistance of the WUO films. In addition, due to having a long alkyl structure for HDI films, it had a lower hardness, higher impact, and bending resistance than IPDI films with a hard and brittle alicyclic structure. However, all of the films showed an excellent adhesion of 10 grades on wood, exhibiting that the WUO coating was suitable for wood finishing. The mass retention of HDI-0.9 was 75.0%, which increased slightly for HDI-Zn and HDI-Ag of 77.4% and 77.1%, respectively. The similar result could also be found in IPDI-films. While, the HDI films had a higher mass retention than IPDI films, it may be due to the molecular weight of HDI (Mw = 10,734) being higher than that of IPDI (Mw = 6126) [21], and the steric effect of IPDI hindering the oxidative polymerization of unsaturated fatty acids in linseed oil [31]. The 60° gloss of HDI-0.9 had a value of 78%, whereas the HDI-Pb showed a decrease to 62% and an HDI-Ag increase to 81%. The results may be attributed to the Zn(HEEP)_2_ and Pb(HEEP)_2_ having a larger particle diameter than nanoAg. Similar results were also shown in IPDI films, but were exhibited at lower gloss values.

The lightfastness of antimicrobial WUO films synthesizing with HDI and IPDI before and after the color fading test are given in Figure 8. The color difference (ΔE*) of WUO films increased with prolonged exposure under ultraviolet light irradiation, accounting for the color changes of the films, but the ΔE* values were found to stabilize at 16 days. The ΔE* values were 6.9, 5.9, and 6.8 for HDI-0.9, HDI-Pb, and HDI-Ag, and were 4.8, 5.5, and 4.9 for IPDI-0.9, IPDI-Zn, and IPDI-Ag, respectively, after ultraviolet light irradiation of 23 days. Generally speaking, it has no significant effect on the lightfastness of WUO films with adding antimicrobial agents. However, the IPDI films had a slight superiority in lightfastness over the HDI films, which all belonged to aliphatic isocyanates.

The Tan δ curves of antimicrobial WUO films synthesizing with HDI and IPDI by a dynamic mechanical analysis (DMA) are given in Figure 9 and Figure 10, respectively. The glass transition temperature (*T*_g_) of HDI-0.9 film, which possessed a softer long alkyl chain, was 10.7 °C and the IPDI-0.9 film, which possessed a harder alicyclic structure, was 55 °C. When adding the nanograde filler of the Ag ion, the *T*_g_ was shifted to a lower temperature, i.e., HDI-Ag was 6.2 °C and IPDI-Ag was 49.3 °C. This was attributed to an interfacial layer of polymer molecules whose chain relaxation dynamics were altered by the interaction with the filler surface. This is accompanied by a shift of the *T*_g_ itself to a lower temperature when the filler surface is organophilic [32]. However, for the micron-grade filler of M(HEEP)_2_, when adding the filler could change the *T*_g_ of the softer polymer such as the HDI film to a higher temperature, i.e., HDI-Pb of 19.8 °C, whereas for the harder polymer such as the IPDI film to a lower temperature, i.e., IPDI-Zn of 39.1 °C [32]. In addition, the IPDI-Zn had a phase separation, this may be due to the Zn(HEEP)_2_ agglutinated mainly in the hard segment of IPDI film located at the main α relaxation, the glass transition *T*_g_ of 39.1 °C, and a second relaxation occurred at −17.9 °C in the unfilled soft segment of the IPDI film.

The TGA curves of antimicrobial WUO films synthesizing with HDI and IPDI are displayed by Figure 11 and Figure 12, respectively. The WUO films with or without antimicrobial agents showed similar behavior of thermal degradation. They consisted of three dominant steps at around 250–350 °C, 350–420 °C, and 420–500 °C, corresponding to the decomposition of the urethane linkage, the degradation of an aliphatic long chain of fatty acid from the linseed oil glyceride component, and the dehydrogenation and depolycondensation of alkyl groups, respectively [33,34,35,36]. The results showed that adding metal-containing antimicrobial agents can slightly enhance the thermal stability of the WUO films, e.g., the weight loss at 450 °C for HDI films were 73%, 69%, and 71% of HDI-0.9, HDI-Pb, and HDI-Ag, respectively, and for IPDI films were 83%, 80%, and 80% of IPDI-0.9, IPDI-Zn, and IPDI-Ag, respectively. The result also confirmed by the studies of Zafar et al. [11] and Chang and Chang [37]. In addition, the M(HEEP)_2_ and nanoAg containing WUO films had no significant difference in thermal stability.

## 4. Conclusions

In the present study, the antimicrobial agents of mono(hydroxyethoxyethyl) phthalate M(HEEP)_2_ (M = Zn, Mn, Pb, and Ca) powders could be readily synthesized and the yields were 43–55%. The results of various antimicrobial agents and dosages on antimicrobial activity of WUO films showed that the WUO film synthesizing with HDI films containing Zn(HEEP)_2_ of 2.0 phr and Pb(HEEP)_2_ of 0.4 phr had the best antibacterial activity for *E. coli* and *S. aureus*, respectively. The IPDI films containing Zn(HEEP)_2_ of 1.0 phr had the best antibacterial activity for both *E. coli* and *S. aureus.* For antifungal activity, the WUO film synthesizing with HDI films containing Pb(HEEP)_2_ of 0.8 phr and Zn(HEEP)_2_ of 2.0 phr as well as IPDI films containing Mn(HEEP)_2_ of 0.2 phr and Zn(HEEP)_2_ of 4.0 phr had the best performances against *G. trabeum* and *L. betulina*, respectively. Comparing with commercial nanoAg powder, the Zn(HEEP)_2_ and Pb(HEEP)_2_ had a superior antifungal efficiency for brown-rot fungus of *G. trabeum* and white-rot fungus of *L. betulina*, while it had a slightly inferior efficiency in antibacterial activity for the Gram-negative bacterium of *E. coli* and Gram-positive bacterium of *S. aureus*. On the properties of WUO films, adding metal-containing antimicrobial agents could slightly enhance the thermal stability, but lowered the gloss of all films, however, the *T*_g_ value increased for the HDI film and decreased for the IPDI film. In addition, they had no significant difference in the film properties including hardness, impact resistance, bending resistance, adhesion, mass retention, and lightfastness between the WUO films with and without adding antimicrobial agents.

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
