# Peer review of "Synthesis and Antimicrobial Activity of Metal-Containing Linseed Oil-Based Waterborne Urethane Oil Wood Coatings"

_polymers, 2020, doi:10.3390/polym12030663_

Round 1

Reviewer 1 Report

The comments for the author attached separately.

Reviewer 2 Report

I have some observations and for this reason I suggest that can be published with major corrections:

In the abstract, and in the rest of the manuscript the authors should be change de name of de acid used, de name correct is “dimethylolpropionic acid”. In the abstract, the authors must clarify the paragraph: “The results also revealed that WUO film synthesizing with HDI films containing 0.2 phr of  Pb(HEEP)2 and IPDI films containing 0.2 phr of Zn(HEEP)2 had the best antimicrobial activity.” In the FTIR spectra, are the curves normalized? And in the Mn(HEEP)2 and Zn(HEEP) the authors should be explained because the carbonyl (C=O) group stretching is so slightly, and the peak in about 1500 cm-1 is very strong. In the NMR spectrometry study the results could be presented in a table, indicating the shift, the integrations and multiplicity for each signal. The same recommendations is for the 13C-NMR spectrum, and please indicate in the espectrum signal, the number for de C assigned. In the spectroscopy analysis, FTIR and RMN, the authors just support your results in reference 11, the authors should look for more references for this parts. In the figure 6, EDS spectra the authors should review the name of elements, that they don't see, change de colour. In the morphological analysis, the authors should reforce with references. In the antibacterial and antifungal activity, how much experiments were make for each system, and how de authors calculated de the statistical values in the tables? Other authors found similar results about this metals, please include more reference. In "Film properties of WUO coatings", the authors attribute the difference in the hardness between two samples to the structure of di-isocianate, other authors found the same results? References are missing in this section. What is the effect of metals on color? How do you explain the influence of metals and the changes observed in this essay? The explanations of the results of “Film properties of WUO coatings” should be made in more detail, and reinforced with bibliography.

Reviewer 3 Report

The films seems to aborb moisture as evident from FTIR data. Authors may also provide datta of water solubility of films as it may determine diffusion during antimicrobial assay.

Round 2

Reviewer 1 Report

Author has tried to improve the manuscript in the light of reviewer’s comments. But still some comments are as follows that author should consider

  1. Abstract

The results showed that the antimicrobial agents of mono(hydroxyethoxyethyl) phthalate M(HEEP)2 (M=Zn, Mn, Pb and Ca)…….. ……….. readily synthesized by FTIR, 1H‐NMR, 13C‐NMR, and EDX identification and the yields were 43~55%. Author should check sentence

  1. The spectral and other characterization of metal-containing linseed oil-based waterborne urethane should be involved in this paper not in future studies. Fillers do not react with the WUO dispersion but they have different electrostatic interactions that effect the performance of the resultant materials.
  2. Preparation of the free standing films and coatings (surface?) for film properties is not clear.
  3. Author should compare the antibacterial and antifungal activity of the synthesized materials with standard drugs. MIC for the systems?
  4. Author should correct typological errors: Zafara should be zafar, Sharmina should be sharmin, Akrama should be Akram, Alame should be Alam

Reviewer 2 Report

In the analysis of spectroscopic results, the authors are requested to reinforce these results with more references, and all the aggregates references are of the same author, they should add references from other authores.

Although the work is original, it has little support for references.

Round 3

Reviewer 1 Report

The author tried to improve the paper, but I cannot accept the paper in the present form. The paper will be accepted if the author will include the spectral and other characterization of linseed oil-based waterborne urethane and metal-containing linseed oil-based waterborne urethane.